# Bariatric Surgery Improves HDL Function Examined by ApoA1 Exchange Rate and Cholesterol Efflux Capacity in Patients with Obesity and Type 2 Diabetes

**DOI:** 10.3390/biom10040551

**Published:** 2020-04-04

**Authors:** Shuhui Wang Lorkowski, Gregory Brubaker, Daniel M. Rotroff, Sangeeta R. Kashyap, Deepak L. Bhatt, Steven E. Nissen, Philip R. Schauer, Ali Aminian, Jonathan D. Smith

**Affiliations:** 1Department of Cardiovascular and Metabolic Sciences, Cleveland Clinic, Cleveland, OH 44195, USA; lorkows@ccf.org (S.W.L.); brubakg@ccf.org (G.B.); 2Department of Quantitative Health Sciences, Cleveland Clinic, Cleveland, OH 44195, USA; rotrofd@ccf.org; 3Endocrinology Institute, Cleveland Clinic, Cleveland, OH 44195, USA; kashyas@ccf.org; 4Bariatric and Metabolic Institute, Department of General Surgery, Cleveland Clinic, Cleveland, OH 44195, USA; Philip.Schauer@pbrc.edu; 5Brigham and Women’s Hospital Heart and Vascular Center and Harvard Medical School, Boston, MA 02115, USA; 6Department of Cardiovascular Medicine, Cleveland Clinic, Cleveland, OH 44195, USA; nissens@ccf.org

**Keywords:** bariatric surgery, HDL function, apoA1 exchange rate, cholesterol efflux capacity

## Abstract

Bariatric surgery improves glycemic control better than medical therapy; however, the effect of bariatric surgery on HDL function is not well characterized. Serum samples were available at baseline, 1-, and 5-years post procedures, for 90 patients with obesity and type 2 diabetes who were randomized to intensive medical therapy (*n* = 20), Roux-en-Y gastric bypass (RYGB, *n* = 37), or sleeve gastrectomy (SG, *n* = 33) as part of the STAMPEDE clinical trial. We examined serum HDL function by two independent assays, apolipoprotein A-1 exchange rate (AER) and cholesterol efflux capacity (CEC). Compared with baseline, AER was significantly higher at 5 years for participants in all treatment groups, but only increased significantly at 1 year in the RYGB and SG groups. CEC was divided into ABCA1-dependent and independent fractions, and the later was correlated with AER. ABCA1-independent CEC increased significantly only at 5 years in both surgical groups, but did not significantly change in the medical therapy group. There was no significant change in ABCA1-dependent CEC in any group. The increase in AER, but not ABCA1-independent CEC, was correlated with the reduction in body mass index and glycated hemoglobin levels among all subjects at 5 years, indicating that AER as a measure of HDL function would be a better reflection of therapy versus CEC.

## 1. Introduction

Bariatric surgery has been demonstrated as an efficient treatment for patients with obesity and type 2 diabetes mellitus by successfully achieving glycemic control in several small randomized clinical trials [1,2,3,4,5]. A recent large matched cohort study further supports the superiority of this treatment showing that metabolic surgery, compared with nonsurgical management, decreases major adverse cardiovascular outcomes in patients with type 2 diabetes and obesity [6].

HDL-cholesterol (HDL-C) levels are inversely associated with cardiovascular disease (CVD) in community based and prospective studies [7,8]. However, recent studies using HDL-C raising drugs, such as niacin and cholesterol ester transfer protein inhibitors, either did not find a reduction in CVD incidence or found reduced CVD incidence that could be attributed to a reduction in non-HDL cholesterol levels [9,10,11,12]. In addition, a genetic Mendelian randomization study found that common genetic variants only associated with HDL-C, and not other lipid traits, were not associated with CVD risk [13]. Therefore, it is thought that it is HDL function rather than HDL-C levels that are protective against CVD [14]. Obesity and type 2 diabetes are associated with lower HDL-C levels, and also with decreased HDL function [15,16,17]. It has been shown that bariatric surgery remarkably improves HDL-C level [1,2,3,4,5]; however, the effect of bariatric surgery on HDL function is still not clear. The current gold standard HDL function assay, the cholesterol efflux capacity (CEC), is a cell-based assay to examine the cholesterol efflux capacity of exogenous acceptors (apoA1 or apoB-depleted serum) from radiolabeled-macrophages (usually RAW264.7 or J774) in which ABCA1 expression can be regulated, allowing the determination of total, ABCA1-dependent, and ABCA1-independent CEC [18]. The CEC of apoB-depleted serum has been used in many studies to investigate the association between CEC and CVD [19,20,21]. 

Roux-en-Y gastric bypass (RYGB) and sleeve gastrectomy (SG) are alternative standard bariatric surgeries [22]. RYGB bypasses most of the stomach by creating a small gastric pouch, a gastrojejunal anastomosis, and a jejunojejunal anastomosis; whereas SG consists of resecting the gastric fundus and most of the gastric body. One study assessed CEC 6 and 12 months after RYGB or SG operations, and, after 12 months, ABCA1-independent CEC increased compared with baseline for both groups, while only increasing in the SG group after 6 months [23]. Another study showed no effect of bariatric surgery on CEC compared with baseline, but used a CEC assay that could not distinguish between ABCA1-dependent versus independent activity [24]; thus, changes in ABCA1-independent CEC might have been obscured. Two additional small studies showed increased total CEC after bariatric surgery [25,26].

We recently published a novel cell-free HDL function assay, the apoA1 exchange rate (AER), by measuring the rate of apoA1 exchange into serum HDL and determined that those in the lowest quartile had significantly higher incidence of major adverse cardiovascular events [27]. In this assay, a doubly labeled apoA1 is incubated with serum and the rate of the exogenous apoA1 incorporation into HDL is determined by a ratiometric fluorescent readout [27]. In the current study, we examined two assays of HDL function, CEC and AER, in patients with obesity and type 2 diabetes who were enrolled in the STAMPEDE clinical trial, which is a randomized, nonblinded, single-center trial to evaluate the efficacy of intensive medical therapy alone versus intensive medical therapy plus RYGB or SG in 150 patients with obesity and type 2 diabetes.

## 2. Materials and Methods

### 2.1. Human Samples

Human serum samples and associated clinical data were collected from the Surgical Treatment and Medications Potentially Eradiate Diabetes Efficiently (STAMPEDE) cohort at Cleveland Clinic (clinicaltrials.gov NCT00432809) [1,2,3]. We studied a subset (*n* = 90) of these subjects for whom serum samples were available at baseline, 1 year, and 5 years after treatment, for a total of 270 serum samples. Serum samples were collected using red top clot activator Vacutainer tubes (BD #367820) and stored in the −80 °C freezer. The subjects had been randomized to receive intensive medical therapy alone (*n* = 20), intensive medical therapy plus RYGB (*n* = 37), and intensive medical therapy plus SG (*n* = 33). All subjects provided written informed consent under protocols approved by the Cleveland Clinic Institutional Review Board. Intensive medical therapy consisted of following the American Diabetes Association guidelines, including lifestyle counseling, weight management, frequent home glucose monitoring, and the use of conventional and newer Food and Drug Administration approved drug therapies (e.g., incretin analogues) [1].

### 2.2. ApoA1 Exchange Indicator

Human apoA1 was incubated with the lipid-sensitive fluorophore NBD-X succinimidyl ester (S1167, Thermo Fisher, Waltham, MA, USA) and the lipid-agnostic fluorophore Alexa Fluor 647 succinimidyl ester (A20006, Thermo Fisher) in 0.1 M sodium bicarbonate for 1 h at room temperature (8:2:1 dye:dye:protein mole ratio). Excess hydroxylamine at pH 8.5 was added to stop the reaction. The conjugate was purified by extensive dialysis against PBS (pH = 7.4). 

### 2.3. ApoA1 Exchange Rate Assay

Five micrograms of the apoA1 exchange indicator (10 μL) was added to 85 μL of PBS in a 96 well plate, then 5 μL of human serum was added to each well. The plate was put into a 96 well fluorescent plate reader set at 37 °C, and NBD (460 nm excitation, 540 nm emission) and Alexa 647 (640 nm excitation, 670 nm emission) fluorescence was measured at 1 min intervals for 1 h. The NBD/Alexa emission ratio increases as the indicator exchanges into HDL in the serum sample. AER was calculated as the slope of linear regression of the NBD/Alexa647 ratio (excluding the non-linear first 10 min). A serum pool from three healthy controls, in triplicate, was included on each plate in order to account for inter-assay variation. The relative AER in each serum sample was normalized to the mean of AER from the pooled controls.

### 2.4. Cellular Cholesterol Efflux Capacity

ApoB-depleted serum was prepared by adding 40 parts of PEG600 to 100 parts of serum, incubating for 20 min at room temperature, and then centrifugation at 10,000× *g* in a microfuge at 4 °C for 30 min. On day 1, RAW264.7 macrophages were plated and cultured in 48-well plates in DMEM with 10% FBS. On day 2, the cells were labeled with 0.5 μCi/mL ^3^H cholesterol in 1% FBS DMEM. On day 3, the cells were treated with or without 0.2 mM 8-Br-cAMP for 16 h to induce ABCA1 expression. On day 4, the cells were washed and chased for 4 h in DMEM with cholesterol acceptor (2.8%, *v*/*v*, apoB-depleted serum, corresponding to 2% serum component). The radioactivity in the chase media was determined after brief centrifugation to pellet debris. Radioactivity in the cells was determined by extraction in hexane:isopropanol (3:2) with the solvent evaporated in a scintillation vial prior to counting. The percent cholesterol efflux was calculated as 100× (medium ^3^H disintegrations per minute (dpm))/(medium dpm + cell dpm). A serum pool from three healthy controls was included on each plate to allow for monitoring of plate to plate and day to day variation. The relative cholesterol efflux capacity was calculated as the cholesterol efflux capacity of each sample divided by the serum pool control.

### 2.5. Statistical Analysis

Descriptive statistics and linear regression analyses were performed using GraphPad Prism software (v.7), San Diego, CA USA. All data was tested for normality. Normally distributed data is shown as mean ± S.D. and was analyzed by students t-test (2 groups) or ANOVA (>2 groups) with Tukey’s posttest. Categorical data was analyzed by the Chi-square test. Effects of all therapies on AER and CEC were performed using matched one-way ANOVA, thus comparing for pooled changes at the three time points among individual subjects. Differences among the three therapies were not analyzed due to differences in sex and HDL-C at baseline. Correlations of changes of AER and ABCA1-independent CEC with changes in glycated hemoglobin (HbA1c), body mass index (BMI), triglycerides (TG), and HDL-C at the 1-year and 5-year time points were performed using Pearson correlation in the statistical language R [28]. Resulting p-values were adjusted for multiple comparisons using a false discovery rate (FDR) approach [29], and results with an FDR *p* < 0.05 were considered statistically significant.

## 3. Results

### HDL Function is Significantly Improved After Bariatric Surgery 

Demographics and clinical characteristics of the 90 subjects at the time of enrollment are shown in Table 1. Female percentage was significantly different among three groups, as was HDL-C levels, with RYGB and SG groups having lower HDL-C than the medical therapy group at baseline (Table 1). 

We assessed HDL function of the 270 serum samples from 90 subjects at three time points by two independent assays, AER and CEC (total CEC, ABCA1-dependent CEC, and ABCA1-independent CEC, respectively). Among total CEC, ABCA1-dependent CEC, and ABCA1-independent CEC, we found that AER was best correlated with the ABCA1-independent cholesterol efflux capacity (r = 0.37, *p* < 0.0001, Appendix A), consistent with our prior results [27]. 

We compared changes in AER and ABCA1-independent CEC in subjects who received intensive medical therapy alone. We found that AER was increased at 5 years versus baseline (*p* < 0.05) but not after 1 year (Figure 1A), while ABCA1-independent CEC was not significantly different at either 1 or 5 years versus baseline (Figure 1B). In addition to the ABCA1-independent CEC, total and ABCA1-dependent CEC for all treatment groups are reported in Appendix A.

For the RYGB group, AER was significantly increased at 1 year (*p* < 0.001, increased by 21.3%) and 5 years (*p* < 0.0001, increased by 18.1%), compared with the baseline levels (Figure 1C). However, ABCA1-independent CEC only shows significant improvement at 5 years (*p* < 0.0001 versus baseline, *p* < 0.01 versus 1 year, Figure 1D).

The HDL function changes over time for the SG group subjects were similar to those observed in the RYGB group. In SG subjects, AER was significantly increased at 1 year (*p* < 0.0001, increased by 22.3%) and 5 years (*p* < 0.0001, increased by 20%) compared with baseline (Figure 1E). Similarly, ABCA1-independent CEC only showed a significant increase at 5 years (*p* < 0.001 versus baseline, *p* < 0.001 versus 1 year, Figure 1F). Therefore, only patients who underwent bariatric operations had significant increases in both AER and ABCA1-independent CEC. Additionally, for the AER measure, the surgical groups had increased HDL function at 1- and 5- years, while the medical therapy group only increased HDL function by year 5.

We examined how the changes in AER and ABCA1-independent CEC correlated with changes in BMI, HbA1c, log TG, and HDL-C at the 1-year and 5-year time points over all subjects in the three treatment groups. At the 1-year time point, we observed a significant inverse correlation of the change in AER with the change in BMI (*p* = 0.0069) and a positive correlation with change in HDL-C levels (*p* = 0.0015, Figure 2). For the ABCA1-independent CEC changes at 1 year, the only significant positive correlation was with the change in log TG (*p* = 0.0015, Figure 3). At the 5-year time point, there were significant inverse correlations for the changes in AER with the changes in BMI (*p* = 0.0069) and HbA1c (*p* = 0.016) and a significant positive correlation with the changes in HDL-C (*p* = 0.001, Figure 4). For the ABCA1-independent CEC changes at 5 years, the only significant positive correlation was with the change in HDL-C (*p* = 0.0058, Figure 5). Thus, the changes in AER appear to capture more of the clinical changes reflecting the status of obesity (changes in BMI) and diabetes (changes in HbA1c) than the ABCA1-independent CEC assay.

We further performed multiple regression analysis of changes of AER or ABCA1-independent CEC with changes in BMI, HbA1c, log TG, and HDL-C at the 1-year and 5-year time points over all subjects in the three treatment groups. At the 1-year time point, there was a significant positive correlation of the change in AER with the change in HDL-C levels (*p* = 0.046, Appendix A). For the ABCA1-independent CEC changes at 1 year, there were significant positive correlations with the change in log TG (*p* < 0.001) and HDL-C levels (*p* = 0.013, Appendix A). At the 5-year time point, we observed a significant inverse correlation of the change in AER with the change in BMI (*p* = 0.041) and a positive correlation with change in HDL-C levels (*p* = 0.006, Appendix A). For the ABCA1-independent CEC changes at 5 years, there were significant positive correlations with the change in log TG (*p* = 0.022) and HDL-C levels (*p* = 0.001, Appendix A). Thus, in the multiple regression model, increases in AER after 5 years were associated with the amount of weight loss (assessed as BMI).

## 4. Discussion

The concept is building that low HDL function, rather than low HDL-C, is mechanistically linked with CVD; therefore, HDL function may be a better target for therapeutic intervention than HDL-C levels [30]. We examined human serum HDL function using AER and CEC assays in individuals with obesity and type 2 diabetes enrolled in the STAMPEDE clinical trial [1,2,3]. This trial compared 1-, 3-, and 5-year outcomes for medical therapy alone or with bariatric surgery (RGBY or SG), using the proportion of patients with a glycated hemoglobin level ≤6% after treatment as the primary end point. Patients started to reach the therapeutic goal of a glycated hemoglobin level of 6.0% or less at the 1-year visit, and the proportions of patients reaching the goal were 12% in intensive medical therapy, 42% in RGBY, and 37% in SG. The results showed that bariatric operations achieved significantly better glycemic control, greater weight loss, and higher HDL-C levels than medical therapy alone in patients with uncontrolled type 2 diabetes at all three time points [1,2,3]. In addition, bariatric surgery, but not medical therapy, was associated with significant decreases (5-year versus baseline) in the inflammatory biomarkers C-reactive protein, myeloperoxidase, and interleukin-6 [31]. The current sub-study found that both RYGB and SG similarly improved HDL function after the surgery, with the AER assay detecting improved HDL function after 1 and 5 years versus baseline, while the CEC assay detected improved function only after 5 years versus baseline. Earlier detection for changes in HDL function, better correlation with clinical markers (changes in BMI and HbA1c), and ease and cost of performance are potential advantages of the AER assay compared with the CEC assay.

Two prior studies assessed the effect of bariatric surgery on HDL function assayed by CEC. Heffron et al., using a CEC assay similar to the one we used, demonstrated that the effect of bariatric surgery on CEC is procedure-dependent at the 6-month time point, with SG, but not RYGB, leading to improved ABCA1-independent CEC in obese, nondiabetic, premenopausal Hispanic women [23]. However, by the 12-month time point, both operations led to improved ABCA1-indepedent CEC versus baseline [23]. Similar to our findings, they did not find any significant changes in ABCA1-dependent CEC at 12 months. Thus, both our study and the study of Heffron et al. found that ABCA1-independent CEC, compared to ABCA1-dependent CEC, was the better indicator of HDL functional changes. This is consistent with our finding that AER and ABCA1-independent CEC are improved after bariatric surgery, and AER is well correlated with ABCA1-independent CEC. Kjellmo et al. found no effect of bariatric surgery on total CEC after 12 months, although a different CEC method was used in this study, without distinguishing ABCA1-dependent and independent CEC [24]. 

Improved HDL function demonstrated by AER and ABCA1-independent CEC after bariatric surgeries could be related to the changes of HDL metabolism. HDL particles are heterogeneous, and obese subjects have significantly decreased larger HDL particles (α1-HDL) compared to lean subjects [32]. While ABCA1-dependent CEC is correlated with lipid-poor smaller HDL particles (pre-β1 HDL) [33], and ABCA1-independent CEC is correlated with lipid-rich larger HDL [34,35]. Therefore, it is possible that bariatric surgeries lead to increases in large HDL particles, and in turn, increased AER and ABCA1-independent CEC. In addition, improved HDL function may also be in part due to a reduction in chronic inflammation after bariatric surgeries.

AER is best correlated with ABCA1-independent CEC both in our previously published study [27] and in the current study. ApoA1 is freely exchangeable between lipid-free and lipid-bound states [36], making AER an indicator of HDL remodeling. The reason for the best association of AER with ABCA1-independent CEC is not known, but may reflect the ability of apoA1 to best exchange onto large lipid-rich HDL that would also be expected to be an excellent acceptor via ABCA1-independent cholesterol efflux. We think the changes of AER and ABCA1-independent CEC at 1- and 5-years are likely to be biologically relevant to the progression of atherosclerosis. However, we could not evaluate the association of AER or ABCA1-independent CEC with CVD events due to insufficient CVD events in this cohort.

Aside from CEC, other HDL functions have been tested in subjects after bariatric surgery. Osto et al. found that RYGB rapidly restores HDL endothelium-protective properties, anti-inflammatory, antiapoptotic, antioxidant effects, and CEC in patients with obesity [37]. Furthermore, adjustable gastric banding has been shown to modify HDL sub-fractions towards atheroprotection by enhancing large HDL and reducing small HDL [25]. The effect of bariatric surgery on HDL function was also assessed in adolescents. It has been shown that SG increases large apoE-rich HDL subspecies, total CEC, and anti-oxidative capacity in adolescent males [26].

This study has some limitations. The STAMPEDE trial assigned patients to intensive medical therapy, RYGB or SG with stratification according to the patients’ use of insulin at baseline [1]. The randomization helps to balance the distribution of confounders among three treatment groups with no difference in clinical parameters at baseline among all 150 subjects; however, serum samples at the three times points were available for only 90 out of 150 participants in the STAMPEDE trial. Due to using a subset of the STAMPEDE subjects, clinical parameters at baseline showed differences in sex percentage and HDL-C level among the three treatment groups in our current study. We analyzed data by comparing HDL function at different time points within each treatment group and performing correlation and multiple linear regression using the changes of HDL function and clinical parameters, which helped avoid possible confounding effects among three treatment groups. Comparisons among the three study arms were not performed.

In conclusion, we demonstrated that bariatric surgery improved CEC and AER functions of HDL, with the AER assay being more sensitive (earlier detection) and much easier to perform vs. the CEC assay. HDL function may be useful as an independent biomarker, which may assess effectiveness of therapeutic risk factor modifications, such as diet and bariatric surgery. 

## Figures and Tables

**Figure 1 biomolecules-10-00551-f001:**
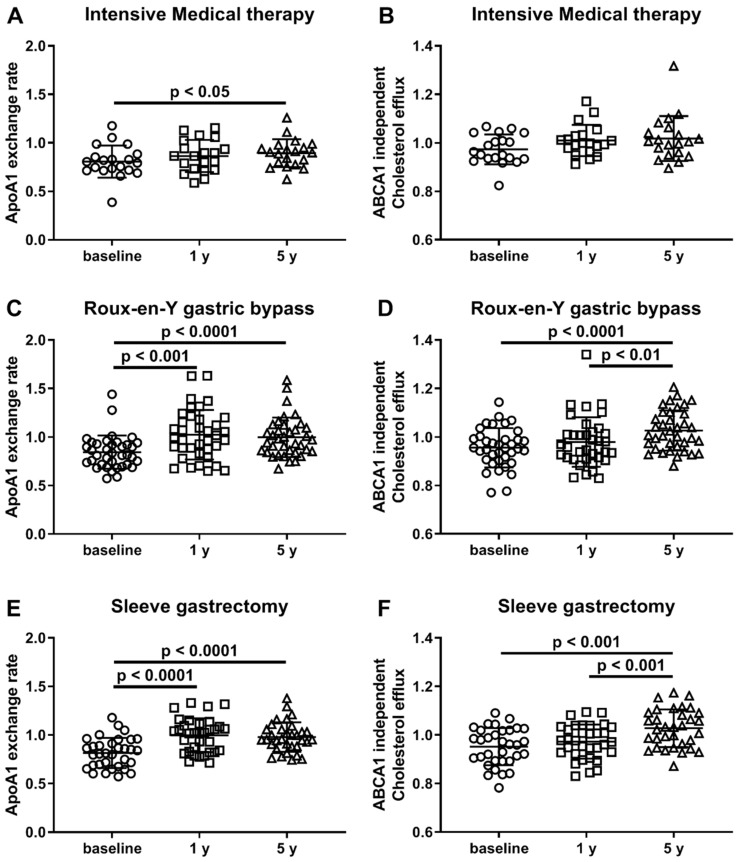
HDL function is improved after bariatric surgery. ApoA1 exchange rate (**A**,**C**,**E**) and ABCA1-independent cholesterol efflux capacity (**B**,**D**,**F**) were assayed in human serum samples from 90 STAMPEDE subjects at baseline (circles), 1 year (squares), 5 years (triangles) after receiving intensive medical therapy (**A**,**B**, *n* = 20), Roux-en-Y gastric bypass (**C**,**D**, *n* = 37) or sleeve gastrectomy (**E**,**F**, *n* = 33) surgeries.

**Figure 2 biomolecules-10-00551-f002:**
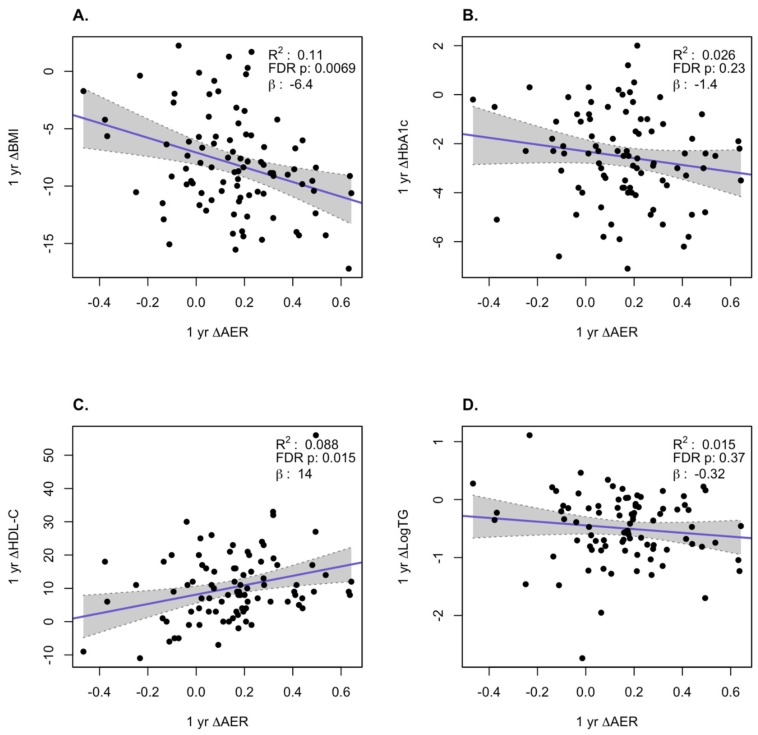
Correlation of changes of apoA1 exchange rate assay (AER) with changes in clinical variables at 1 year. Changes in BMI (**A**), HbA1c (**B**), HDL-C (**C**), and log TG (**D**). The gray region is the 95% confidence interval.

**Figure 3 biomolecules-10-00551-f003:**
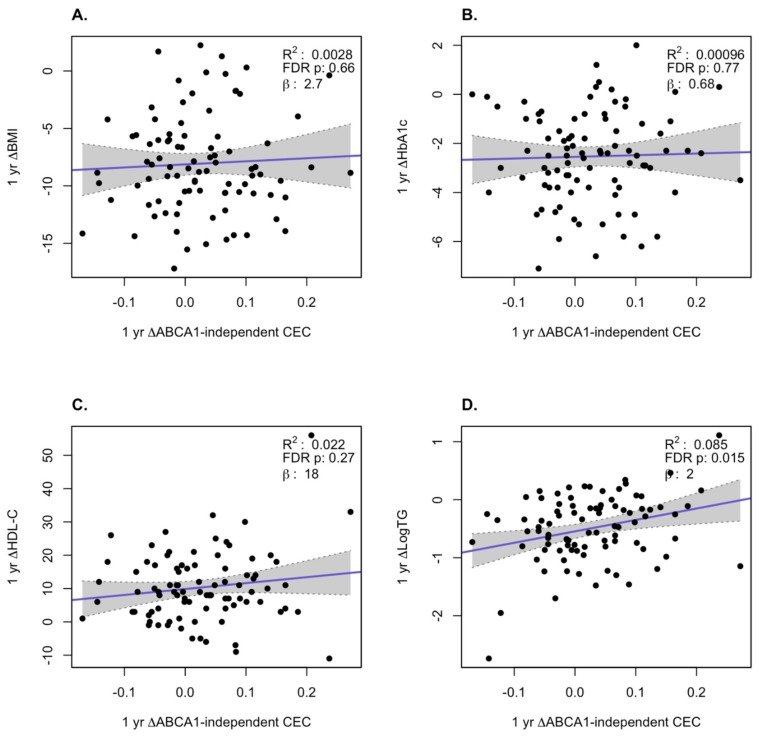
Correlation of changes of ABCA1-independent CEC with changes in clinical variables at 1 year. Changes in BMI (**A**), HbA1c (**B**), HDL-C (**C**), and log TG (**D**). The gray region is the 95% confidence interval.

**Figure 4 biomolecules-10-00551-f004:**
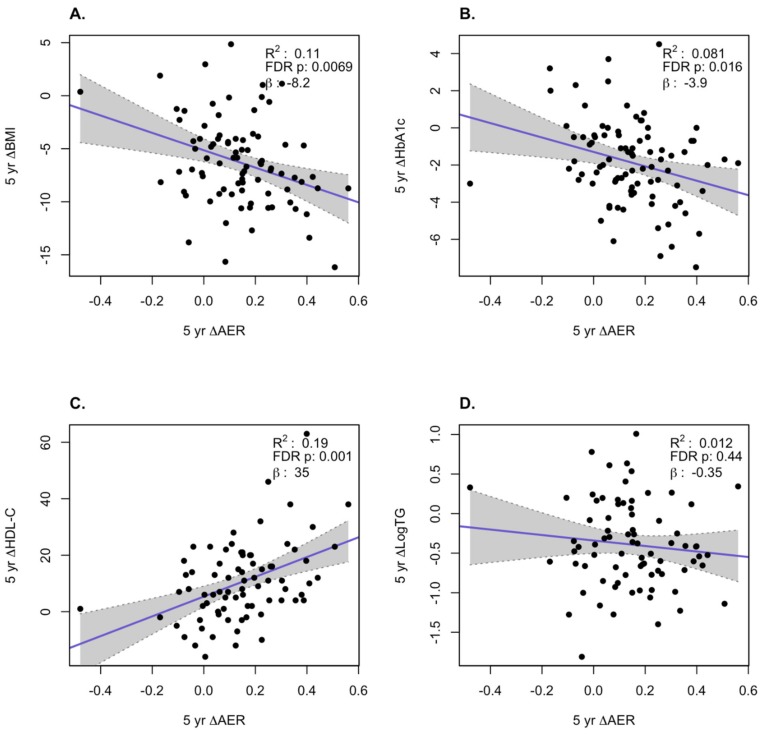
Correlation of changes of apoA1 exchange rate assay (AER) with changes in clinical variables at 5 years. Changes in BMI (**A**), HbA1c (**B**), HDL-C (**C**), and log TG (**D**). The gray region is the 95% confidence interval.

**Figure 5 biomolecules-10-00551-f005:**
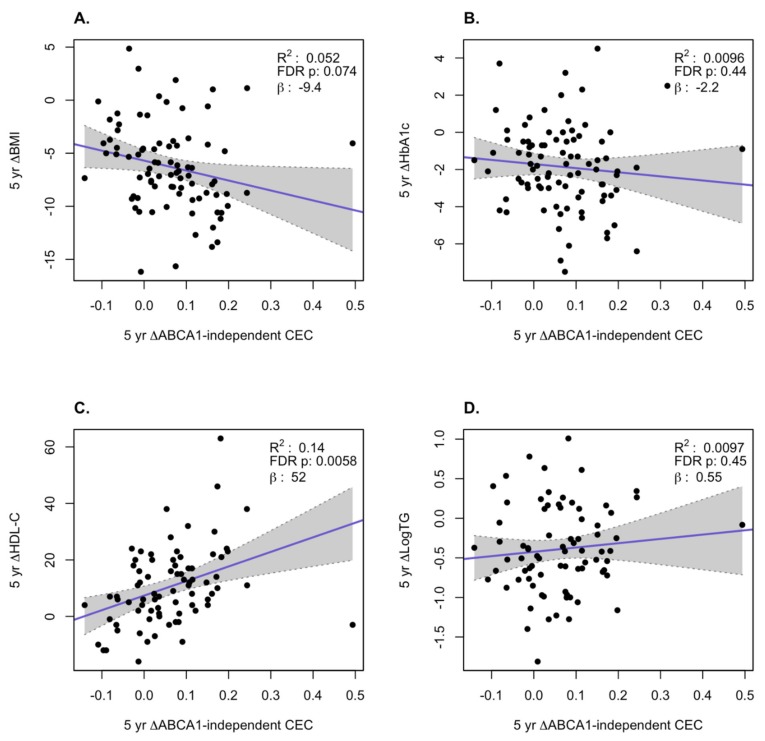
Correlation of changes of ABCA1-independent CEC with changes in clinical variables at 5 years. Changes in BMI (**A**), HbA1c (**B**), HDL-C (**C**), and log TG (**D**). The gray region is the 95% confidence interval.

**Table 1 biomolecules-10-00551-t001:** Baseline characteristics of the STAMPEDE cohort.

Characteristic	Medical Therapy (N = 20)	Roux-en-Y Gastric Bypass (N = 37)	Sleeve Gastrectomy (N = 33)	*p*-Value
Duration of diabetes (yr)	10.15 ± 5.75	7.60 ± 5.22	8.61 ± 4.38	0.20
Glycated hemoglobin (%)	8.81 ± 0.98	9.18 ± 1.37	9.69 ± 1.89	0.11
Use of insulin (%)	50.0	45.9	54.5	0.77
Age (yr)	52.58 ± 5.46	48.52 ± 8.43	48.54 ± 7.43	0.11
Female (%)	70.0	54.1	81.8	0.045
Body Mass Index (kg/m^2^)	35.84 ± 2.92	37.01 ± 3.32	35.87 ± 4.32	0.34
Body weight (kg)	102.1 ± 10.94	107.3 ± 15.91	99.28 ± 17.56	0.10
Waist (cm)	112.7 ± 6.93	117.0 ± 9.51	113.5 ± 10.98	0.18
Waist-to-hip ratio	0.95 ± 0.07	0.97 ± 0.07	0.95 ± 0.08	0.39
White race (%)	70.0	75.7	72.7	0.89
Smoking (%)	20.0	35.1	15.2	0.13
Metabolic Syndrome (%)	85.0	97.3	93.9	0.20
Hypertension (%)	75.0	75.7	60.6	0.33
Dyslipidemia (%)	85.0	89.2	84.8	0.84
HDL-C (mg/dl)	51.50 ± 12.45	44.16 ± 11.64	41.91 ± 9.10	0.0094

Values are mean ± SD, or %. *p* values were calculated by ANOVA test for continuous data and Chi-square test for categorical factors.

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
