# Peer review of "Bariatric Surgery Improves HDL Function Examined by ApoA1 Exchange Rate and Cholesterol Efflux Capacity in Patients with Obesity and Type 2 Diabetes"

_biomolecules, 2020, doi:10.3390/biom10040551_

Round 1

Reviewer 1 Report

Shuhui Wang Lorkowski et al. investigated serum HDL function by apolipopritein A-1 exchange rate (AER) and cholesterol efflux capacity (CEC) in the patients who intensive medical therapy gastric bypass and sleeve gastrectomy. AER is significantly higher at 5 years and ABCA1-independent CEC increased at 5 years in both surgical groups. The increase in AER but not ABCA-1-independnt CEC was correlated with the reduction in BMI and HbA1c levels.

Major comments

  1. In page 4 lines 128-132, the authors should explain why did AER well-correlated with ABCA1-independent CEC in introduction or discussion sections.
  2. The authors should present odds ratios for CVD events by AER or ABCA1-independent CEC using logistic regression analysis.
  3. The changes of AER and ABCA1-independent CEC at 1-year and 5-year are biologically relevant in the vascular function and progression of atherosclerosis? The changes of AER and ABCA1-independent CEC seemed to be marginal.
  4. The authors should perform multiple regression analysis using AER or ABCA1-independent CEC as dependent variables, and various clinical parameters as independent variables.

Author Response

Major comments

  1. In page 4 lines 128-132, the authors should explain why did AER well-correlated with ABCA1-independent CEC in introduction or discussion sections.

We added the following to the Discussion to address this issue: “AER is best correlated with ABCA1-independent CEC both in our previously published study [27] and in the current study. ApoA1 is freely exchangeable between lipid-free and lipid-bound states [36], making AER an indicator of HDL remodeling. The reason for the best association of AER with ABCA1-independent CEC is not known, but may reflect the ability of apoA1 to best exchange onto large lipid-rich HDL that would also be expected to be an excellent acceptor via ABCA1-independent cholesterol efflux.”

We also added in the Discussion: “Improved HDL function demonstrated by AER and ABCA1-independent CEC after bariatric surgeries could be related to the changes of HDL metabolism. HDL particles are heterogeneous, and obese subjects have significantly decreased larger HDL particles (α1-HDL) compared to lean subjects [32]. While ABCA1-dependent CEC is correlated with lipid-poor smaller HDL particles (pre-β1 HDL) [33], ABCA1-independent CEC is correlated with lipid-rich larger HDL [34, 35]. Therefore, it is possible that bariatric surgeries lead to increases in large HDL particles, and in turn, increases AER and ABCA1-independent CEC.”

  1. The authors should present odds ratios for CVD events by AER or ABCA1-independent CEC using logistic regression analysis.

There were only 2 subjects in the entire cohort with subsequent MACE, so the sample size was insufficient to perform logistic regression analysis. We added in the Discussion: “However, we could not evaluate the association of AER or ABCA1-independent CEC with CVD events due to insufficient CVD events in this cohort.”

  1. The changes of AER and ABCA1-independent CEC at 1-year and 5-year are biologically relevant in the vascular function and progression of atherosclerosis? The changes of AER and ABCA1-independent CEC seemed to be marginal.

We added in the Discussion: “We think the changes of AER and ABCA1-independent CEC at 1-year and 5-year are likely to be biologically relevant in the progression of atherosclerosis. However, we could not evaluate the association of AER or ABCA1-independent CEC with CVD events due to insufficient CVD events in this cohort.” Levels of AER and ABCA1-independent CEC were significantly increased compared to baseline levels (i.e. AER had a range of 18.1%-22.3% increase at both 1 year and 5 years), therefore, we don’t think the changes of AER and ABCA1-independent CEC to be marginal.

  1. The authors should perform multiple regression analysis using AER or ABCA1-independent CEC as dependent variables, and various clinical parameters as independent variables.

We performed multiple regression analysis of changes of AER or ABCA1-independent CEC with changes in clinical variables at 1 year and 5 years (Table S1-S4). We added in the Results: “We further performed multiple regression analysis of changes of AER or ABCA1-independent CEC with changes in BMI, HbA1c, log TG, and HDL-C at the 1-year and 5-year time points over all subjects in the three treatment groups. At the 1-year time point, there was a significant positive correlation of the change in AER with the change in HDL-C levels (p=0.046, Supplemental Table S1). For the ABCA1-independent CEC changes at 1 year, there were significant positive correlations with the change in Log TG (p<0.001) and HDL-C levels (p=0.013, Supplemental Table S2). At the 5-year time point, we observed a significant inverse correlation of the change in AER with the change in BMI (p=0.041) and a positive correlation with change in HDL-C levels (p=0.006, Supplemental Table S3). For the ABCA1-independent CEC changes at 5 years, there were significant positive correlations with the change in Log TG (p=0.022) and HDL-C levels (p=0.001, Supplemental Table S4). Thus, in the multiple regression model, increases in AER after 5 years were associated with the amount of weight loss (assessed as BMI).”

Reviewer 2 Report

In the present manuscript, Lorkowski and collaborators investigated the effects of bariatric surgery on HDL function. The authors examined a total of 90 patients from which they collected serum (baseline, one and 5 years after the interventions) and analyzed HDL function via two independent methods: apolipoprotein A-1 exchange rate (AER) and cholesterol efflux capacity (CEC). Even though the authors report very interesting and relevant findings, there are some issues (grouped as minor and major) that need to be addressed before the paper can be recommended for publication.

MINOR:

Line 20-21: Please inform your audience that 1 and 5 years are “post” procedures.

Lines 70-72: Please provided more details about how the serum samples were collected and stored before the experiments (tube used, how the serum was obtained, storage temperature, etc.).

Line 82: “extensive dialysis” against which buffer (pH)?

Lines 103-104: Please define dpm (disintegrations per minute).

Fonts in the references do not match the fonts used throughout the main text of the manuscript.

MAJOR:

General comments on plots (Fig.1 and S1-S3): When the authors make the data overlay, please make sure to choose “hollow” and not “solid” symbols such that it is possible for the reader to see the mean and standard deviation clearly on the plot. Don’t forget to mention the n for each group on the figure captions.

General comment on the results/ discussion: Even though the findings are interesting and relevant, a mechanism that explains why bariatric surgery improves HDL function is lacking. Why do the authors think that bariatric surgery is improving HDL function and other clinical parameters? The improvements in BMI and glycated hemoglobin, e.g., are a consequence of the bariatric surgery or a consequence of HDL function improvement? Please elaborate a bit more on that.

Lines 51-56: A little bit more background (with references) should be given to an audience that is not familiar with the “gold standard HDL functional assay”. The same applies to the surgical procedures: Roux-en-Y gastric bypass (RYGB) or sleeve gastrectomy (SG). A brief explanation about how these procedures differ and why one is preferred for a specific patient over another would be appreciated by an audience that is not familiar with the surgical theater.

Lines 57-58: “Another study showed no effect of bariatric surgery on CEC compared with baseline, but used a CEC assay that could not distinguish between ABCA1-dependent versus independent activity”. Make sure to tell your audience why it is important to distinguish between ABCA1-dependent versus independent activity in the context of the assay you performed.

Line 66: Please explain briefly what the STAMPEDE clinical trial is.

Line 73: Please describe what the “intensive medical therapy” consists of.

Lines 130-132: All the correlations that the authors show in Fig S1 are poor since the data points are clearly not linearly distributed. Despite this, the authors claim that “AER was best correlated with the ABCA1-130 independent cholesterol efflux capacity”. What is the importance/ meaning of this result given that the correlation is poor? Please, elaborate.

Author Response

MINOR:

Line 20-21: Please inform your audience that 1 and 5 years are “post” procedures.

We revised the sentence to “Serum samples were available at baseline, 1-, and 5-years post procedures”.

Lines 70-72: Please provided more details about how the serum samples were collected and stored before the experiments (tube used, how the serum was obtained, storage temperature, etc.).

We added in the Materials and Methods: “Serum samples were collected using red top clot activator Vacutainer tubes (BD #367820) and stored in the -80C freezer.”

Line 82: “extensive dialysis” against which buffer (pH)?

We revised the sentence to “The conjugate was purified by extensive dialysis against PBS (pH = 7.4)”.

Lines 103-104: Please define dpm (disintegrations per minute).

We edited this section to read: “The percent cholesterol efflux was calculated as 100 × (medium 3H disintegrations per minute (dpm)) / (medium dpm + cell dpm).”

Fonts in the references do not match the fonts used throughout the main text of the manuscript.

We made the font change for the references to match the main text.

MAJOR:

General comments on plots (Fig.1 and S1-S3): When the authors make the data overlay, please make sure to choose “hollow” and not “solid” symbols such that it is possible for the reader to see the mean and standard deviation clearly on the plot. Don’t forget to mention the n for each group on the figure captions.

We have changed solid symbols into hollow ones in Figures 1 and S1-S3, and we added the n for each group in figure captions.

General comment on the results/ discussion: Even though the findings are interesting and relevant, a mechanism that explains why bariatric surgery improves HDL function is lacking. Why do the authors think that bariatric surgery is improving HDL function and other clinical parameters? The improvements in BMI and glycated hemoglobin, e.g., are a consequence of the bariatric surgery or a consequence of HDL function improvement? Please elaborate a bit more on that.

We added in the Discussion: “Improved HDL function demonstrated by AER and ABCA1-independent CEC after bariatric surgeries could be related to the changes of HDL metabolism. HDL particles are heterogeneous, and obese subjects have significantly decreased larger HDL particles (α1-HDL) compared to lean subjects [32]. While ABCA1-dependent CEC is correlated with lipid-poor smaller HDL particles (pre-β1 HDL) [33], ABCA1-independent CEC is correlated with lipid-rich larger HDL [34, 35]. Therefore, it is possible that bariatric surgeries lead to increases in large HDL particles, and in turn, increases AER and ABCA1-independent CEC. In addition, improved HDL function may also be in part due to a reduction in chronic inflammation after bariatric surgeries.”

Lines 51-56: A little bit more background (with references) should be given to an audience that is not familiar with the “gold standard HDL functional assay”. The same applies to the surgical procedures: Roux-en-Y gastric bypass (RYGB) or sleeve gastrectomy (SG). A brief explanation about how these procedures differ and why one is preferred for a specific patient over another would be appreciated by an audience that is not familiar with the surgical theater.

We added in the Introduction: “The current gold standard HDL functional assay, cholesterol efflux capacity (CEC) assay is a cell-based assay to examine the cholesterol efflux capacity of a exogenous acceptors (apoA1 or apoB-depleted serum) from radiolabeled-macrophages (usually RAW264.7 or J774) in which ABCA1 expression can be regulated, allowing the determination of total, ABCA1-dependent, and ABCA1-independnet CEC [18].”.

We also added “Roux-en-Y gastric bypass (RYGB) and sleeve gastrectomy (SG) are alternative standard bariatric surgeries [22]. RYGB bypasses most of the stomach by creating a small gastric pouch, a gastrojejunal anastomosis, and a jejunojejunal anastomosis; whereas sleeve gastrectomy (SG) consists of the resecting the gastric fundus and most of the gastric body.”

Lines 57-58: “Another study showed no effect of bariatric surgery on CEC compared with baseline, but used a CEC assay that could not distinguish between ABCA1-dependent versus independent activity”. Make sure to tell your audience why it is important to distinguish between ABCA1-dependent versus independent activity in the context of the assay you performed.

We edited the Introduction: “Another study showed no effect of bariatric surgery on CEC compared with baseline, but used a CEC assay that could not distinguish between ABCA1-dependent versus independent activity [24]; thus, changes in ABCA1-independent CEC might have been obscured.”

In the Discussion, we further write about why ABCA1-independent CEC may be best correlated with AER. “AER is best correlated with ABCA1-independent CEC both in our previously published study [27] and in the current study. ApoA1 is freely exchangeable between lipid-free and lipid-bound states [36], making AER an indicator of HDL remodeling. The reason for the best association of AER with ABCA1-independent CEC is not known, but may reflect the ability of apoA1 to best exchange onto large lipid-rich HDL that would also be expected to be an excellent acceptor via ABCA1-independent cholesterol efflux.”

Line 66: Please explain briefly what the STAMPEDE clinical trial is.

We added in the Introduction: “In the current study, we examined two assays of HDL function, CEC and AER, in patients with obesity and type 2 diabetes mellitus who were enrolled in the STAMPEDE clinical trial, which is a randomized, nonblinded, single-center trial to evaluate the efficacy of intensive medical therapy alone vs, intensive medical therapy plus RYGB or SG in 150 patients with obesity and type 2 diabetes mellitus.”

Line 73: Please describe what the “intensive medical therapy” consists of.

We added in the Materials and Methods: “Intensive medical therapy consisted of following the American Diabetes Association guidelines, including lifestyle counseling, weight management, frequent home glucose monitoring, and the use of conventional and newer Food and Drug Administration approved drug therapies (e.g. incretin analogues) [1].”

Lines 130-132: All the correlations that the authors show in Fig S1 are poor since the data points are clearly not linearly distributed. Despite this, the authors claim that “AER was best correlated with the ABCA1-130 independent cholesterol efflux capacity”. What is the importance/ meaning of this result given that the correlation is poor? Please, elaborate.

Among the total CEC, ABCA1-dependent CEC, and ABCA1-independent CEC, we observed ABCA1-independent CEC is best correlated with AER, and the correlation is statistically significant; therefore, we think it is fair to say with the minor revision to the sentence, “Among total CEC, ABCA1-dependent CEC, and ABCA1-independent CEC, we found that AER was best correlated with the ABCA1-independent cholesterol efflux capacity (r=0.37, p<0.0001, Supplemental Figure S1), consistent with our prior results [27].”

Reviewer 3 Report

This is a very well written manuscript from Lorkowski et. al. who have used samples from a well characterized clinical trial cohort to show that HDL function is perhaps a better indicator/factor in obesity-associated CVD patients. The topic and findings are significant. I have a few questions/suggestions/comments that might help the readers to understand this work better:

  1. The description of STAMPEDE study needs to be made clearer. For example, where did the HbA1c levels start to reach 6 or below after treatment?
  2. Do the authors think that the presumed variability of clinical parameters prior to treatment have no confounding effect on their study outcomes for this manuscript? Something the authors might want to discuss.
  3. It would be appropriate for the authors to discuss the gender skew in MT and SG groups, compared to RYGB.

Author Response

This is a very well written manuscript from Lorkowski et. al. who have used samples from a well characterized clinical trial cohort to show that HDL function is perhaps a better indicator/factor in obesity-associated CVD patients. The topic and findings are significant. I have a few questions/suggestions/comments that might help the readers to understand this work better:

  1. The description of STAMPEDE study needs to be made clearer. For example, where did the HbA1c levels start to reach 6 or below after treatment?

We added in the Introduction: “In the current study, we examined two assays of HDL function, CEC and AER, in patients with obesity and type 2 diabetes mellitus who were enrolled in the STAMPEDE clinical trial, which is a randomized, nonblinded, single-center trial to evaluate the efficacy of intensive medical therapy alone vs, intensive medical therapy plus RYGB or SG in 150 patients with obesity and type 2 diabetes mellitus.” We also added the baseline HbA1c levels in Table 1. Finally, we added in the Discussion: “Patients started to reach the therapeutic goal of a glycated hemoglobin level of 6.0% or less at the 1-year visit, and the proportions of patients reaching the goal were 12% in intensive medical therapy, 42% in RGBY, and 37% in SG.”

  1. Do the authors think that the presumed variability of clinical parameters prior to treatment have no confounding effect on their study outcomes for this manuscript? Something the authors might want to discuss.

We added in the Discussion: “STAMPEDE trial assigned patients to intensive medical therapy, RYGB or SG with stratification according to the patients’ use of insulin at baseline [1]. The randomization helps to balance the distribution of confounders among three treatment groups with no difference in clinical parameters at baseline among all 150 subjects; however, serum samples at the three times points were available for only 90 out of 150 participants in the STAMPEDE trial. Due to using a subset of the STAMPEDE subjects, clinical parameters at baseline showed differences in sex percentage and HDL-C level among the three treatment groups in our current study. We analyzed data by comparing HDL function at different time points within each treatment group and performing correlation and multiple linear regression using the changes of HDL function and clinical parameters, which helped avoid possible confounding effects among three treatment groups.”

  1. It would be appropriate for the authors to discuss the gender skew in MT and SG groups, compared to RYGB.

We added in the Discussion: “Due to using a subset of the STAMPEDE subjects, clinical parameters at baseline showed differences in sex percentage and HDL-C level among the three treatment groups in our current study.”

Round 2

Reviewer 2 Report

Since the authors carefully addressed my concerns and improved the manuscript considerably I can recommend the current version for publication.